# Perioperative Risk Factors for Prolonged Blood Loss and Drainage Fluid Secretion after Breast Reconstruction

**DOI:** 10.3390/jcm11030808

**Published:** 2022-02-03

**Authors:** Tonatiuh Flores, Florian J. Jaklin, Alexander Rohrbacher, Klaus F. Schrögendorfer, Konstantin D. Bergmeister

**Affiliations:** 1Department of Plastic Surgery, University Hospital of St. Poelten, Karl Landsteiner University of Health Sciences, Dr.-Karl-Dorrek-Straße 30, 3500 Krems, Austria; florian.jaklin@meduniwien.ac.at (F.J.J.); alexander@rohrbacher.at (A.R.); klaus.schroegendorfer@stpoelten.lknoe.at (K.F.S.); kbergmeister@gmail.com (K.D.B.); 2Clinical Department of Plastic, Aesthetic and Reconstructive Surgery, University Clinic of St. Poelten, 3100 St. Poelten, Austria; 3Clinical Laboratory for Bionic Extremity Reconstruction, Department of Plastic, Reconstructive and Aesthetic Surgery, Medical University of Vienna, 1090 Vienna, Austria

**Keywords:** breast cancer, mastectomy, postoperative management, anemia, blood loss, anticoagulation, axillary dissection

## Abstract

Background: Surgical breast reconstruction is an integral part of cancer treatment but must not compromise oncological safety. Patient-dependent risk factors (smoking, BMI, etc.) are said to influence perioperative outcomes and have often been investigated. Here, we analyzed independent perioperative risk factors for increased postoperative blood loss or drainage fluid volume loss and their possible impact. Methods: Patients undergoing breast reconstructions after breast cancer with either tissue expanders, definitive breast implants, or autologous breast reconstruction were analyzed. The collected data on patients’ characteristics, blood, and drainage fluid loss were correlated and statistically investigated. Results: Traditional patient-dependent risk factors did not influence blood loss or drainage volumes. On the contrary, patients with preoperative anemia had significantly higher drainage outputs compared to non-anemic patients (U = 2448.5; *p* = 0.0012). The administration of low molecular weight heparin showed a tendency of increased drainage output. Similar correlations could be seen in prolonged procedure time, all of which contributed to prolonged hospital stay (τb = 0.371; *p* < 0.00001). Conclusions: Preoperative anemia is one of the most critical factors influencing postoperative drainage fluid output. Previously assumed patient-dependent risk factors did not affect drainage output. Preoperative anemia must be monitored, and if possible, treated preoperatively to reduce postoperative morbidity.

## 1. Introduction

Breast cancer is the most common cancer among women in industrial countries [1,2,3,4]. Although substantial progress in diagnosis and treatment has been achieved, the complete removal of the breast gland via mastectomy is still often necessary. Without reconstruction, these procedures often leave women physiologically scarred, leading to dire psychological consequences. Due to the high curative rate of breast cancer and overall long-term survival rate, these days, women without breast reconstruction suffer longer from the effects of mastectomy [4]. Reconstructive breast surgery has therefore become an integral part in cancer treatment but must not compromise oncological safety. In line, state-of-the-art guidelines demand patients being offered a reconstructive consultation to restore the female appearance and reduce the postoperative impact on women’s health [1,2,3]. However, patients often require neoadjuvant radiation or chemotherapy after tumor removal and reconstruction, which necessitates complete wound healing. Therefore, identifying the perioperative risk factors for prolonged wound healing or hospital stay are of great importance to potentially optimize the postoperative course and minimize the above risks. Some of these aspects are oncology-related (e.g., mastectomy type, or axillary dissection) and cannot be influenced, whereas others are not and may be subject to optimization [5,6,7,8,9,10,11].

Persistent blood loss and increased drainage fluid volumes have been identified as autonomous risk factors for adversely affecting short- and long-term outcomes, as well as postoperative mortality [12,13,14,15,16,17,18,19,20]. Despite the low incidence of postoperative complications after breast reconstructions, our patients often require prolonged hospitalization due to increased drainage output or because of increased (yet not life-threatening) postoperative blood loss [21]. Since the placement of (wound-) drainages (tubes installed within the wound pocket to drain accumulated fluids) is crucial to prevent postoperative seroma formation or to monitor possible secondary bleedings (all of which could compromise surgical outcome), analyzing the amount of fluid promoted daily is essential in breast reconstruction [22,23,24,25]. Several studies have been conducted investigating patient-related factors influencing postoperative drainage fluid volumes and the associated impact on patient’s quality of life (QoL) [23,25,26,27]. However, few studies have undertaken the prediction of postoperative drainage fluid output, and consecutively prolonged in-hospital stay, analyzing independent risk factors [25,28].

As breast cancer treatment is very well defined by state-of-the-art guidelines, several risk factors (such as postoperative radiation therapy) cannot be modified to increase patient safety. However, other factors may very well be accessible to optimization and not interfere with the oncological treatment. Preoperative anemia has been identified as significantly increasing patient’s 30-day morbidity and length of hospital stay (LOS), but to our knowledge, a direct association to postoperative drainage volumes has yet not been done [19]. Hence, in this study, we aimed to focus on analyzing drainage fluid volume and perioperative blood loss as patient independent risk factors for prolonged hospitalization. Here, we investigated three patient cohorts undergoing breast reconstruction after breast cancer and risk factors influencing postoperative blood loss and drainage fluid volume.

## 2. Materials and Methods

In this study, we analyzed breast reconstructions after breast cancer performed at the Clinical Department for Plastic, Aesthetic, and Reconstructive Surgery at the University Hospital St. Poelten, between January 2010 and December 2020. The study was conducted as a single center retrospective review of prospectively acquired data. Ethical approval was obtained from the local institutional review board at the Karl Landsteiner University of Health Sciences Krems (reference number: 1044/2020).

The target population consisted of all adult female patients (≥18 years) who had reconstructive breast surgery following mastectomy due to breast cancer. Patients were divided into three groups based on the type of reconstruction (tissue expander, definitive breast implants, or autologous breast reconstruction). Breast reconstructions with tissue expanders were performed as delayed-immediate reconstruction. Data acquisition was achieved through the electronical patient files.

Factors reviewed included patients’ age, BMI, smoking status, perioperative antithrombotic prophylaxis, perioperative hemoglobin (Hb), operation time, operation side (unilateral or bilateral), drainage volume, and axillary dissection. Hemoglobin values were analyzed prior to surgery and on the first postoperative day until patient discharge. Anemia is defined as values below 12 g/dL in women classified by the WHO [29]. Criterion for patients receiving blood transfusions at our department was a hemoglobin level below 12 g/dL with clinical signs of hypoperfusion (dizziness, hypotension, tachycardia, abnormal fatigue, etc.). Drainage output was documented every 12 h until the removal of drainage catheters. Drainage removal was typically conducted if the output was less than 30 mL in 24 h. Patients were not discharged from hospital until (among others) drainage removal.

### 2.1. Perioperative Management

All patients received intravenous antibiotic shielding prior to surgery with either 1.5 g of Cefuroxime (Curocef^®^, GlaxoSmithKline Manufacturing S.p.A., Verona, Italy), 3 g of an Ampicillin/Sulbactam combination (Unasyn^®^, Pharma Latina S.r.l., Latina (LT), Italy), or in case of penicillin allergy 600 mg of Clindamycin (Dalacin^®^, Fareva Amboise, Pocé-sur-Cisse, France). An antibiotic application was administered at least 30 min before surgical incision.

Patients received postoperative antithrombotic prophylaxis by the subcutaneous injection of either Enoxaparin or Dalteparin; the injection was given at the earliest six hours after surgery. Dosages were set at a prophylactic level based on the patients’ weight (0.5 mg of LMWH x the patient’s weight). In case of prior thromboembolic events or preoperatively established antithrombotic therapy, low molecular heparin dosages were bridged accordingly (1 mg of LMWH × the patient’s weight).

### 2.2. Operative Procedure

Mastectomy types included nipple-sparing mastectomy (NSM), skin sparing mastectomy (SSM), modified radical mastectomy (MRM), and simple mastectomy. If the sentinel lymph node tested positive, axillary dissection was performed simultaneously. Free tissue flaps included DIEP-flaps (deep inferior epigastric perforator flap) and TMG-flaps (transverse myocutanous gracilis flap). All expanders were inserted submuscular (subpectoral) and had textured and anatomical properties (Mentor^®^ Becker™-Expander). Pocket preparation was performed through incising the major pectoral muscle parallel to its muscle fiber course approximately at the level of the fourth to fifth intercostal space. The serratus anterior fascia was partially raised to support the implant inferiorly and laterally. In case of tissue expander-based reconstruction, port systems were installed at the level of the anterior axillary line. If no axillary dissection was performed, subcutaneous and submuscular drainages were placed. In the case of axillary dissection, one additional drainage was inserted in the axillary wound cavity.

### 2.3. Statistics and Data Management

The primary end point was to investigate the impact of perioperative factors (anemia, procedure time, anticoagulants, hospitalization time) on postoperative blood and fluid loss as well as the differences of such between the three groups. Data collection and processing were performed with Microsoft Excel (Microsoft corp., Washington, DC, USA), and statistical analyses were performed using SPSS version 26.0.0.0 (IBM Corp., New York, NY, USA). For metric variables, median, interquartile range (IQR) and range are reported. Testing for statistical significance of the differences between fluid loss regarding implant-based and autologous tissue reconstruction was performed with the Mann-Whitney-U-Test, since the initial analysis of normality using the Shapiro-Wilk-test did not confirm normal distribution in the subgroups. A contingency analysis was performed using the Kendall-tau-b test (τb), to determine whether correlations between the duration of the surgical procedures and the volume of drainage output exist. Additionally, analyses using the Kruskal–Wallis test were performed. Two-sided *p* ≤ 0.05 was regarded as statistically significant.

## 3. Results

### 3.1. Patient Demographics

In total, 359 breast reconstructions due to breast cancer in 297 patients were identified during the study period. Thereof, 41 were breast reconstructions after mastectomy in non-breast-cancer patients (e.g., persistent mastectomies with failed conservative treatment) and were therefore excluded. Additionally, 24 breast reconstructions were excluded due to a combination of multiple reconstruction procedures (implant and lipofilling, implant and pedicled flap). From the remaining 294 breast reconstructions, 37 were excluded due to missing data. Consequently, 257 breast reconstructions in 195 patients met our criteria and were included in our study.

Of the 257 breast reconstructions in 195 patients, 133 (51.8%) were performed unilaterally and 124 (48.2%) were performed bilaterally. Breast reconstruction via tissue expander was performed in 82 (42.1%) out of 195 patients, via definitive breast implant in 64 (32.8%) patients and in 49 (25.1%) patients via autologous breast reconstruction. Thereof, 32 were deep inferior epigastric perforator (DIEP) flaps and 17 were transverse myocutaneous gracilis (TMG) flaps. In 75 (38.5%) patients, breast reconstruction was performed immediately. Thereof, 56 (28.8%) were direct to implant, and 19 (9.7%) were direct to free tissue flap (15 (7.7%) DIEP-flaps, 4 (2%) TMG-flaps). The remaining 120 (61.5%) patients experienced delayed-immediate reconstruction. Here, eight (4.1%) were tissue expander exchange to definitive implant, 30 (14.4%) were tissue expander exchange to free tissue flap, and 82 (42%) were sole tissue expander implantation. In our population group, 28 (14.3%) patients received preoperative chemotherapy; eight of these patients (4.1%) were bilateral.

Patients’ median age was similarly distributed with 49.1 years (IQR 15.4) in the tissue expander group, 46.8 years (IQR 10.9) in the implant group, and 49.7 years (IQR 13.9) in the autologous tissue group (Table 1). Inclusion criteria of patients suitable for breast reconstruction were women aged between 18–99 years, total mastectomy, skin-sparing mastectomy, nipple-sparing mastectomy, implant-based breast reconstruction (silicone gel implant or tissue expander), and breast reconstruction with autologous tissue (e.g., DIEP, TMG, TRAM).

The median BMI was 24.5 kg/m^2^ (IQR 6.9) in tissue expander patients, 23.4 kg/m^2^ (IQR 6.7) in definitive implant patients, and highest in autologous tissue patients with 26.9 kg/m^2^ (IQR 6.1). In total, 16% of patients were active smokers, with 14% in the breast implant group, 17.5% in the expander group, and 13% in the free tissue flap group. Of 23 preoperatively anemic patients (hb lower than 12 g/dL), four (7.5%) were from the tissue expander group, 14 (18.9%) from the definitive implant group, and five (11.1%) from the free tissue group. As none of these patients showed critical preoperative anemia (hb lower than 8.5 g/dL), surgery was performed on all patients.

### 3.2. Surgical Procedure Time

The median operation time was 2.7 h (IQR 1.2) in the implant group, 2.2 h (IQR 1.8) in the tissue expander group, and 8.4 h (IQR 3.3) in the free tissue flap group. Overall median operation time was 2.9 h (IQR 3.5). Women reconstructed with autologous tissue or definitive breast implants showed a weak, yet not statistically significant correlation for hemoglobin levels and procedure time, displayed in slightly decreased hemoglobin levels following longer surgery time (definitive implants: τ_b_ = 0.099, *p* = 0.414; autologous tissue reconstruction: τ_b_ = 0.147, *p* = 0.170). Patients reconstructed with tissue expanders showed a high and statistically significant correlation for lower hemoglobin levels following longer surgeries (τ_b_ = 0.201, *p* = 0.029) (Figure 1).

Overall, an increase in surgery duration did not correlate with an increase in blood loss. Furthermore, no meaningful correlation was observed in the length of surgery and postoperative drainage loss (τ_b_ = 0.187; *p* < 0.0001). However, women reconstructed with silicone implants or tissue expanders both displayed strong significant correlation of surgery time and postoperative drainage loss (tissue expanders: τ_b_ = 0.553; *p* < 0.0001; implants: τ_b_ = 0.337; *p* < 0.0001) (Figure 2).

### 3.3. Blood and Fluid Loss

Postoperative hemoglobin loss in the first 24 h was 2.7 g/dL (IQR 1.7) in all groups. Patients showed a median decrease in hemoglobin of 2.2 g/dL (IQR 1.6) in the tissue expander group and 2.4 g/dL (IQR 2.0) in the definitive implant group. Patients who underwent autologous tissue reconstruction had a median hemoglobin drop of 3.3 g/dL (IQR 1.4). Differences of hemoglobin loss differed significantly between the groups (Chi-square = 20.1; *p* < 0.0001; df = 2). (Figure 3 and Figure 4).

Preoperatively, anemia was present in 23 (13.4%) patients. Patients with preoperative anemia were observed to have statistically significant higher drainage outputs in comparison to non-anemic patients (U = 2448.5; *p* = 0.0012) (Figure 5).

Overall, no statistically significant correlation between blood loss and drainage output was present. Only within the groups could a weak correlation between drainage output and decreased hemoglobin levels be observed (Figure 6).

Reconstruction via definitive implants showed a higher median drainage output in contrast to autologous tissue-based reconstruction (Figure 7). This effect was not present for expanders.

Compared to patients with one dose of low molecular heparin, patients with double anticoagulation showed a non-significant tendency for higher (2.7 g/dL (1 dosage), vs. 3.4 g/dL (2 dosages), *p* = 0.080601) postoperative blood loss (Figure 8). This effect was seen regardless of the type of low molecular anticoagulant given (Fragmin or Dalteparin). The effect of the number of anticoagulation doses did not affect drainage output. Furthermore, there was no correlation between the cumulative dose of heparin and hemoglobin decrease.

### 3.4. Hospitalization

The median hospitalization time was eight days (IQR 4.3) in the tissue expander group, 8.5 days (IQR 4) for definitive breast implant patients, and eleven days (IQR 3) for free tissue reconstructed patients. The median overall in-hospital stay was nine days (IQR 4.3). A strong statistical correlation of elevated postoperative fluid volume output with longer hospital stays has been observed among our patients (τ_b_ = 0.371; *p* < 0.00001). Overall, patients with higher loss of hemoglobin required statistically longer in-patient treatment (τ_b_ = 0.183; *p* = 0.003). This was especially seen in tissue expander reconstructed patients (τ_b_ = 0.27; *p* = 0.005).

### 3.5. Mastectomy and Axillary Dissection

Overall, 147 patients underwent a mastectomy, 69 of which were in the expander group, 59 in the definitive breast implant group, and 19 in the autologous tissue transfer group. Patients who experienced axillary dissection showed significantly higher drainage fluid output in our findings (643 mL overall median fluid output in axillary dissection group vs. 300 mL overall median fluid output in non-axillary dissection group, U = 1926; *p* < 0.00001).

If no mastectomy was performed (in case of exchange from expander to definitive implant/free tissue flap, or exchange from definitive implant to expander/free tissue flap), the drainage fluid volume output was still increased in tissue expander and free autologous tissue-based reconstructions. In patients reconstructed with definitive breast implants, the drainage fluid volume was significantly lower in case of no mastectomy (in case of exchange of definitive implant to expander/free tissue flap) (138 mL overall median fluid output less: Chi2(2) = 8.578, *p* = 0.014). Besides, correlation in the decrease in hemoglobin levels could be observed in patients with additional axillary dissection. No correlation was seen in the pre- and post-operative hemoglobin levels regarding mastectomy types.

## 4. Discussion

Breast reconstruction is vital in rehabilitating women with breast cancer into daily life [4]. It is an integral part of state-of-the-art cancer treatment and must not compromise oncological safety. These days, many breast reconstructions are performed immediately following cancer removal. Procedures vary in extent based on breast cancer type, hormonal status, and affected lymph nodes. These independent factors can negatively influence the postoperative healing course, as has been previously published [6]. In the case of mastectomy and axillary dissection, postoperative drainage fluid loss and blood loss are significantly higher, and patients require longer in-patient treatment; resulting in longer convalescence before further treatment can be initiated [20,30]. As treatment schemes are based on the breast cancer’s characteristics, reconstruction procedures are sometimes limited, and therefore, many factors influencing surgical outcomes cannot be altered due to oncologic safety. Therefore, this study investigated independent factors influencing blood or drainage fluid loss and their impact on postoperative in-hospital stay.

Besides investigating independent risk factors for breast reconstruction, patient-based risk factors (age, smoking, BMI, LMWH dosages) were also statistically evaluated via a grouped correlation analysis. Here, we analyzed factors possibly influencing drainage volume output. Statistics showed *p*-values > 0.05 (age *p* = 0.573, smoking *p* = 0.079, BMI *p* = 0.35, LMWH dosages *p* = 0.821). Our findings indicate that common risk factors such as age or weight did not affect blood or drainage fluid loss and did not lead to prolonged hospital treatment.

This fortifies current scientific evidence that breast reconstruction is generally safe in patients deemed fit for surgery. However, in our practice, none of the above factors are an absolute contraindication for any type of breast reconstruction [31]. However, we advise that these factors may increase the likeliness of complications, which must be considered when choosing the reconstruction procedure.

In our analyses, the key aspect for prolonged in-patient treatment were constant and increased drainage volumes, prohibiting their removal. While it may be common practice to transfer patients with closed-suction drains in place to an outpatient setting, this is uncommon in many European countries. To the best of our knowledge, most European insurance companies cover expenses for prolonged hospitalization, even if patient discharge is only due to prolonged suction drainages in place. Therefore, identifying independent risk factors for prolonged drainage fluid output, and reducing such beforehand, may help to reduce in-patient treatment and associated negative effects on patients.

In our analyses, patients with preoperative anemia (hemoglobin-levels under 12 g/dL) showed significantly elevated postoperative drainage fluid output, regardless of reconstruction type. While there are many hypotheses about this pathophysiological process, it may be a consequence of suboptimal blood clotting [32]. Unfortunately, anemia in breast cancer patients is frequently seen due to the tumor disease or subsequent chemotherapy. Therefore, patients should be screened for preoperative anemia, and if possible, optimized preoperatively [22,23,24,25]. This is especially relevant, as the rate of anemia increased from approximately 12.5% preoperatively to 78.2% postoperatively in the entire patient population.

Despite the high rate of postoperative anemia, only few patients had clinical symptoms and thus required blood transfusions. In general, we try to avoid blood transfusions considering the recent discoveries about the negative effects on tumor progression [33,34]. Given that the difference in hemoglobin loss was significant but small, we also did not change our practice in offering autologous breast reconstruction to patients with (partially) decreased blood levels. However, if profound, we do consider immediate-delayed reconstructions by expander and secondary reconstruction with autologous breast reconstruction a viable and often beneficiary option. Given the high rate of postoperative need for radiation, this is often our generally preferred reconstructive approach that many centers use [35,36,37].

Regarding drainage fluid loss, we encountered elevated drainage fluid volumes in the tissue expander and definitive implant group in our analyses compared to free flaps in primary reconstructions (Figure 7). Although it is believed that breast implants (as a foreign material) lead to the irritation of the surrounding tissue and subsequent higher secretion, this effect was not seen when comparing only patients without mastectomy or axillary dissection in implant- or expander-based reconstructions. Thus, elevated drainage fluid volume was predominantly an effect of the mastectomy in this cohort, as this was not seen in secondary reconstruction with tissue expanders or definitive implants. In secondary reconstruction with implants or expanders, only preoperatively anemic patients showed overall elevated drainage fluid volumes compared to free flaps. This again highlights the significance of anemia and its effect on prolonged drainage output. Therefore, we believe that the presence of definitive implants or tissue expanders does not increase drainage fluid output per se but is rather mistakenly attributed instead of anemia or mastectomy. Thus, we conclude that significantly increased postoperative drainage fluid output, regardless of reconstruction modality, is a direct result of anemia, as seen in patients with preoperative hemoglobin below 12g/dL.

Anticoagulation has also been thought as a risk factor for blood loss or high drainage fluid volumes. In this study, no significant differences were evident. However, a small tendency was seen towards higher drainage volumes in patients receiving LMWH twice a day. Given that most breast cancer patients need sufficient thromboembolism prophylaxis as indicated by Caprini scores, these assessment scores may often not be individualized enough [38,39,40]. However, it highlights the need to critically evaluate sufficient but not overly aggressive anticoagulation.

The results of this study may be limited due to the size of the analyzed population and its retrospective nature. However, similar studies did not have greater populations and showed similar results [41].

Based on our results, we perform screenings of all patients prior to reconstruction for anemia and refer affected individuals to specialists for preoperative counseling. In addition, we assess these patients critically for the proper doses of anticoagulation to prevent thromboembolic events and at the same time minimize negative effects.

## 5. Conclusions

Increased blood loss and drainage fluid output lead to longer in-hospital treatment. Previously believed factors influencing postoperative blood loss and drainage fluid output were not responsible for this in our study. However, preoperative anemia was the most important influencing factor, and patients with preoperative anemia showed significantly increased drainage volumes. Improving these risk factors by preoperative treatment may thus help to reduce the duration of drainages in place, their volume output, and consequently, in-hospital stay. As patients receiving perioperative antithrombotic prophylaxis show elevated, yet not significantly increased drainage volumes, the necessity of its administration should also be meticulously evaluated. Overall, breast reconstruction is a safe treatment to help patients recover from breast cancer, and perioperative optimization may help to reduce unwanted side effects.

## Figures and Tables

**Figure 1 jcm-11-00808-f001:**
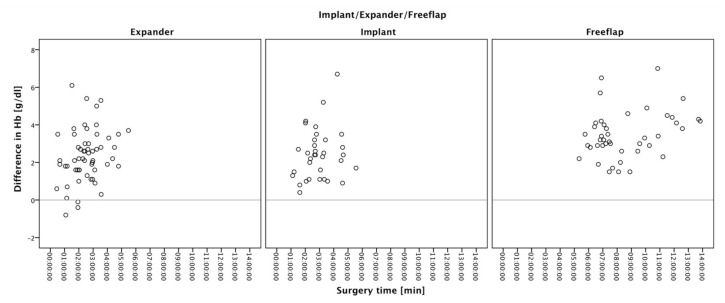
Scatter plot correlation of surgery duration and hemoglobin loss: patients undergoing autologous tissue reconstruction or reconstruction with definitive breast implants showed a weak (but not significant) correlation of hemoglobin loss and procedure time (autologous tissue reconstruction: τ_b_ = 0.147, *p* = 0.170; definitive implants: τ_b_ = 0.099, *p* = 0.414). Patients reconstructed with tissue expanders showed a significantly increased postoperative hemoglobin loss correlating with surgery duration (tissue expander: τ_b_ = 0.201, *p* = 0.029).

**Figure 2 jcm-11-00808-f002:**
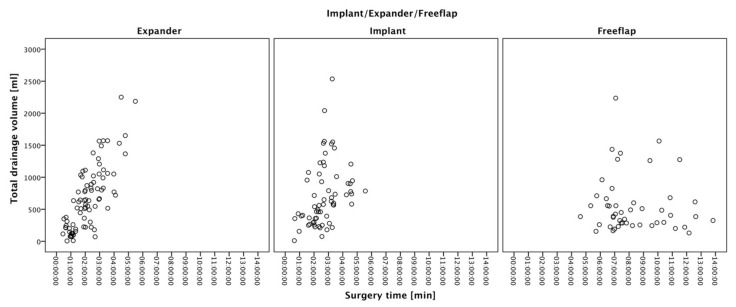
Scatter plot correlation of postoperative drainage fluid and surgery duration: overall, no significant correlation between total drainage output and surgery time was seen in our groups (τ_b_ = 0.187; *p* < 0.0001). However, patients reconstructed with tissue expander or definitive breast implants showed a significant correlation between higher drainage output and surgery time (tissue expander: τ_b_ = 0.553, *p* < 0.0001; definitive implants: τ_b_ = 0.337, *p* < 0.0001).

**Figure 3 jcm-11-00808-f003:**
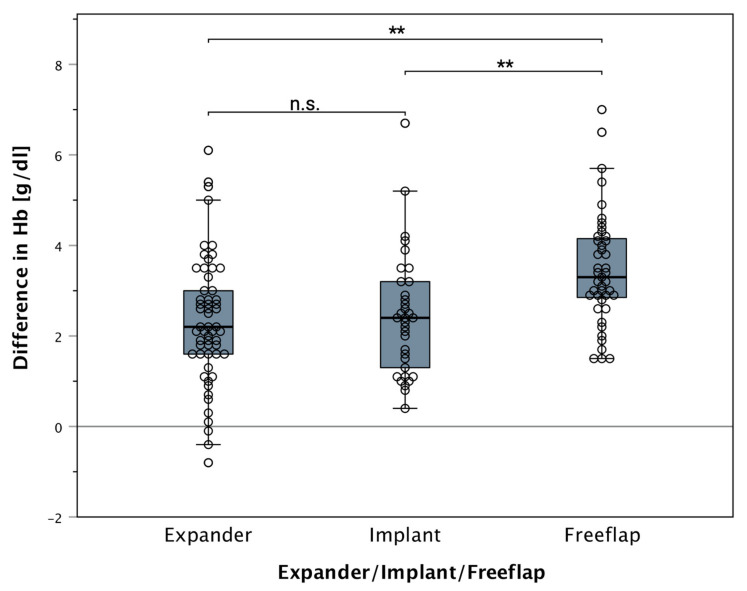
Boxplot of pre- and post-operative hemoglobin difference and used reconstruction method: patients reconstructed with autologous tissue showed a statistically significant drop in hemoglobin levels postoperatively (free tissue flap: 3.3 g/dL). Prosthetic-based reconstructions showed no significant hemoglobin decrease. (Median: tissue expander: 2.2 g/dL, definitive implant: 2.4 g/dL). “n.s.” indicating no significance. The stars (**) indicating a statistically significant difference between the groups (*p* < 0.05). All data points (including outliers) can be seen as scattered dots overlying the boxplot.

**Figure 4 jcm-11-00808-f004:**
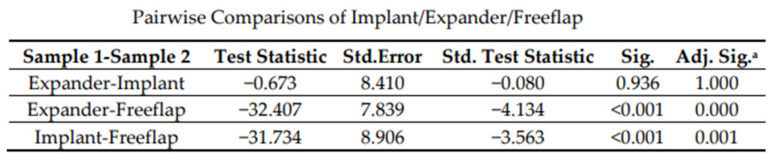
Pairwise comparison of implant, expander, free flap: each row tests the hypothesis that the level of blood loss between our groups are the same. A statistical significance can be seen in the comparison between “Expander-Free flap” and “Implant-Free flap”, showing that patients reconstructed with autologous tissue showed a significant drop in postoperative hemoglobin levels (significance level is 0.05).

**Figure 5 jcm-11-00808-f005:**
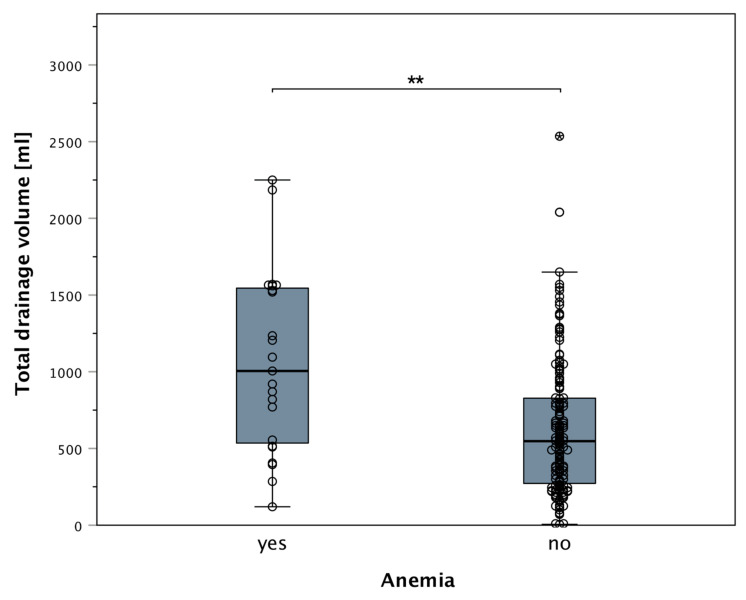
Boxplot of patients with preoperative anemia and drainage fluid volume: patients with preoperative anemia displayed significantly higher drainage output compared to preoperatively non-anemic patients (U = 2448.5; * *p* = 0.0012). The stars (**) indicate a statistically significant difference in preoperative anemic versus preoperative not anemic patients regarding postoperative total drainage volume (*p* < 0.05). All data points (including outliers) can be seen as scattered dots overlying the boxplot.

**Figure 6 jcm-11-00808-f006:**
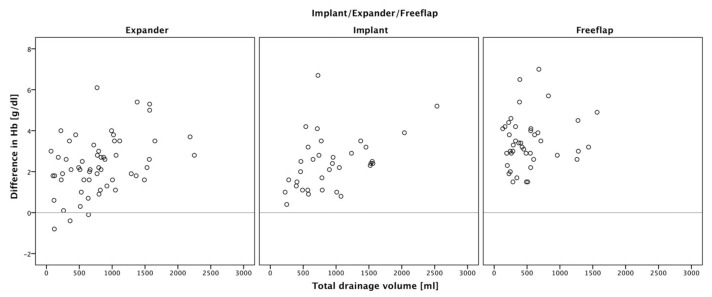
Scatter plot correlation analysis of hemoglobin levels and drainage fluid output within the groups: no significant correlation of increased drainage fluid output could be observed in our patients (tissue expander: τ_b_ = 0.26, *p* = 0.00434, N = 57; definitive implants: τ_b_ = 0.28, *p* = 0.01979, N = 34; free tissue flaps: τ_b_ = 0.08, *p* = 0.474, N = 42). Only within the tissue expander and definitive implant group a weak correlation could be seen. Simultaneously, hemoglobin level decrease only appeared to be minimal in these two groups.

**Figure 7 jcm-11-00808-f007:**
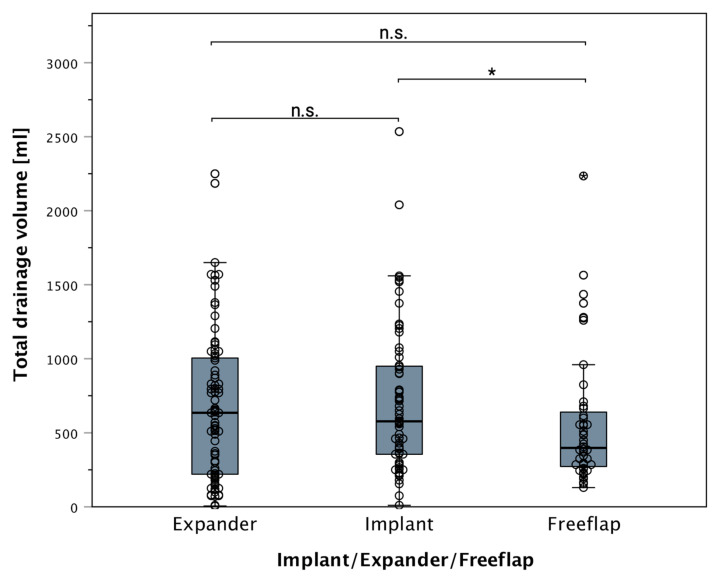
Boxplot of drainage fluid output between groups at reconstruction: a significant difference in the drainage fluid volume is present between definitive implant-based reconstructions and the free tissue flap group (in primary reconstruction) (* *p* = 0.045). This effect is not significant in secondary reconstructions without mastectomy. Only if patients were preoperatively anemic, a significant increase in drainage fluid output is seen at secondary reconstruction. Implicating anemia as being the cause of elevated drainage fluid output (regardless of reconstruction modality). “n.s.” indicating no significance. The star (*) indicating a statistically significant difference between the groups (*p* < 0.05). All data points (including outliers) can be seen incorporated as scattered dots.

**Figure 8 jcm-11-00808-f008:**
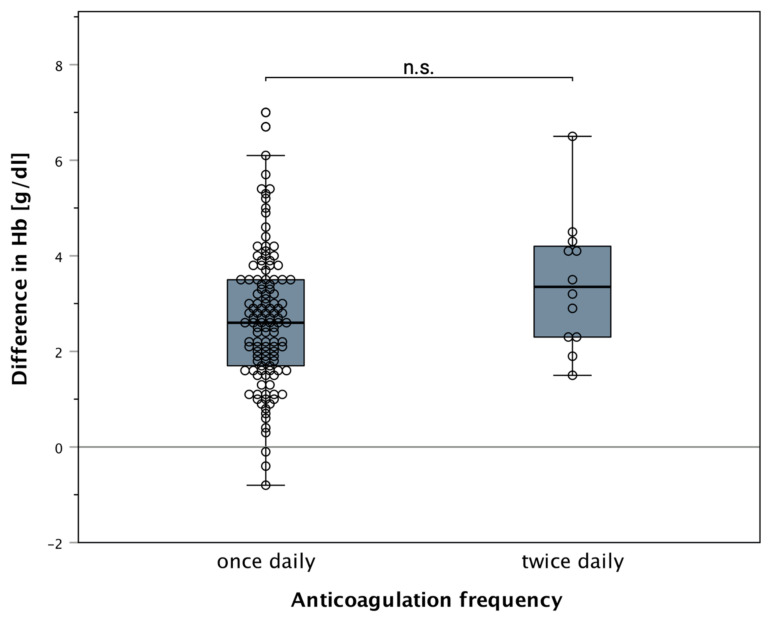
Boxplot correlation of hemoglobin drop regarding frequency of low molecular heparin application: a non-significant tendency of increased drainage fluid output can be observed between patients receiving single- or double-dosages of low molecular weight heparin (2.7 g/dL (1 dosage), vs. 3.4 g/dL (2 dosages), *p* = 0.080601). The type of heparin used had no effect on such. “n.s.” indicating no significance. All data points (including outliers) can be seen incorporated as scattered dots.

**Table 1 jcm-11-00808-t001:** Baseline characteristics of patients: significantly higher drainage volumes were seen in prosthetic-based breast reconstruction (577.5 mL in definitive implant patients, 635 mL in expander patients versus 397.5 mL in autologous tissue reconstructed patients-drainage fluid variation was seen as being an effect of anemia). Overall, 23 (13.4%) patients showed preoperative anemia.

Patient Characteristic	Implant	Expander	Free Flap	Overall Patients
Number	64	82	49	195
Age-years				
Median	46.8	49.1	49.7	48.5
Min-Max (Range)	19.8–73.2	29.5–79.9	29.4–66.7	19.8–79.9 (60.1)
IQR	10.9	15.4	13.9	13.5
Operated breasts-No. (%)	97 (37.7%)	104 (40.5%)	56 (21.8%)	257
Duration of surgery-hours (±mastectomy)				
Median (IQR)	2.7 (1.2)	2.2 (1.8)	8.4 (3.3)	2.9 (3.5)
Volume Drains-mL				
Median (IQR)	577.5 (597.5)	635 (788.8)	397.5 (386.25)	552.5 (646.3)
Hb Difference-g/dL				
Median (IQR)	2.4 (2)	2.2 (1.6)	3.3 (1.4)	2.7 (1.7)
Mastectomy				
No. (%)	59 (92.2%)	69 (84.1%)	19 (38.8%)	147 (75.4%)
Median weight [g] (IQR)	406 (380)	522.5 (454.5)	700 (510)	520 (425)
Anemia preoperative-No. (%)	4 (7.5%)	14 (18.9%)	5 (11.1%)	23 (13.4%)
Anemia postoperative-No. (%)	31 (73.8%)	45 (73.8%)	40 (87%)	116 (77.9%)
Low molecular weight heparin-doses (%)				
Once daily	62 (96.9%)	79 (96.3%)	36 (73.5%)	177 (90.8%)
Twice daily	1 (1.6%)	0 (0%)	13 (26.5%)	14 (7.2%)
None	1 (1.6%)	3 (3.7%)	0 (0%)	4 (2.1%)
Axillary dissection-No. (%)	29 (45.3%)	51 (64.6%)	9 (37.5%)	89 (53.3%)
Length of stay-days				
Median (IQR)	8.5 (4.0)	8 (4.3)	11 (3.0)	9 (4.3)

## Data Availability

All the data analyzed during the current study are available from the corresponding author on reasonable request.

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
