# Peer review of "Perioperative Risk Factors for Prolonged Blood Loss and Drainage Fluid Secretion after Breast Reconstruction"

_jcm, 2022, doi:10.3390/jcm11030808_

Round 1
Reviewer 1 Report
The manuscript seems overall suitable for publication in the present special issue. It makes sense that preoperative anemia represents a risk factor for increased drainage output. However this has not been demonstrated as clearly yet. The data and statistics are very well curated and analyzed. The discussion and message are relatable and do not leave much room for further interpretation. I congratulate the authors to their findings and well exectued research.
Author Response
Dear Reviewer 1,
Thank you very much for your comment and your praise. We tried our best as to state our findings properly and as to support it with proper statistics. We are pleased to hear, that you consider the quality of our paper sufficient for a publication at the Journal of Clinical Medicine. We are very interested in further investigation and improving breast reconstruction after breast cancer. We hope that our paper will help other physicians to further improve surgical short- and long-term outcome and to support women even further while battling this severe disease.
Kind regards,
Tonatiuh Flores, M.D.
Reviewer 2 Report
In this accepted for review manuscript the authors have evaluated perioperative risk factors for prolonged blood loss and drain-fluid secretion after breast reconstruction.
The Authors took up the important and current clinical issue. They convincingly emphasize practical value and clinical benefits deriving from their modus operandi described in the manuscript.
In terms of formal requirements, the concept of the article is compliant. The authors have employed a method which complies with the criteria to be met by scientific papers.
In order to increase the value of its content, I suggest the following:
Materials and methods
- The test period should be specified
- Whether the study concerned immediately or immediately and delayed breast reconstructions – requires clarification
- The number of cases of preoperative chemotherapy should be determined
- Line 88 - the use of low molecular heparin dosages should be clarified
Results
- In which cases were patients with anemia operated on
- Line 121 - It is best to present a graphic diagram of the qualification of patients for the reconstruction procedure
- Line 246 (and 249 and Table no. 1) – “If no mastectomy was performed” - breast reconstruction without mastectomy? – requires clarification
Discussion
- “As breast cancer characteristics propter treatment schemes, many factors cannot be altered due to oncologic safety” – This sentence requires revision.
Author Response
Dear Reviewer 2,
Thank you very much for your comments, which helped us to improve the manuscript significantly. Please find our answers below:
- Thank you very much for drawing attention to the fact, that the test period has not been added to our paper. The test period has been specified and added to our paper (line 75).
- We are very pleased, that you pointed out the missing clarification about the time of reconstruction. Our study concerned immediate, as well as delayed-immediate reconstructions. A clarification about the time of reconstruction has been added (line 84, line 240).
- Thank you very much reviewer 1, for this comment. We checked our data and included your annotation. The number of patients who received preoperative chemotherapy has been added (line 243)
- Thank you for the comment on line 88 (“the use of low molecular heparin dosages should be clarified”). The dosages of low molecular weight heparin administered to the patients included in our study have been added.
- (line 104-109)
Results:
- Thank you very much for your question. All patients with preoperative anemia received surgery, as none of the patients showed critical anemia. The cases of patients operated on with anemia, has been added (line 176).
- Thank you for your advice, as to include the Indications for patients who experienced breast reconstruction at our department. The indications have been added to our paper (line 165).
- Thank you for the comment. You are right as to mention, that the term “no mastectomy performed” seems confusing. A clarification about “breast reconstruction without mastectomy” has been put in brackets. We refer to “breast reconstruction without mastectomy” to the second stage in delayed-immediate reconstruction, meaning the exchange of the tissue expander after adequate expansion of the skin envelop and further reconstruction (definitive implant or free tissue flap) (line 298).
Discussion:
- Referring to the sentence “As breast cancer characteristics proper treatment […]” (line 308): The sentence has been revised critically and changed to “As treatment schemes are based on the breast cancer’s characteristics, reconstruction procedures are sometimes limited, and therefore many factors influencing surgical outcomes cannot be altered due to oncologic safety.“
We thank you very much for your time and effort as to support us in improving our findings. We hope that you find our corrections sufficient for a publication at the Journal of Clinical Medicine. If you are not pleased with our corrections, we would be honored as to hearing from you, as we are eager to better our skills in publishing, and to help other physicians in reducing the patients’ burden after breast reconstruction.
Kind regards,
Tonatiuh Flores, M.D.
Reviewer 3 Report
The authors have put forward an analysis of certain risk factors that may worsen patient outcomes following breast reconstruction after breast cancer surgery. While the results are sound and the conclusions important in a clinical context, the authors need to visualize and present the data better to support their claims. The introduction also needs to be fleshed out and supported with references to previous works in this area. I hope the authors can sufficiently answer my comments and concerns about this work:
- Please sufficiently introduce the risk factors you are studying in the introduction. The results focus mostly on the type of reconstruction and a couple of other factors such as anemia and dose of anticoagulants, but not enough justification is provided as to why these factors are singled out. There must be significant justification for choosing these factors, as well as a summarization of the literature which have looked at these factors before.
- The authors mention the term drainage volume at multiple points, but an explanation of that term will be beneficial to someone not familiar with mastectomy and breast reconstruction. This can be added in the Introduction.
- Line 127 – Can the authors explain what they mean by “sufficient data quality”? It will help to include the metrics that were used to assess the quality of the data, especially since these data points were excluded from later analyses. E.g. were these outliers or were there serious concerns about how the data was collected? If it’s the former, it will be prudent to include the data in the graphs, while providing justification on why they weren’t included in statistical analyses. If it’s the latter then can the authors provide some examples of what the concerns were?
- Figure 3,4,6,7 - The way the graphs are plotted is a little confusing. I assume that the data points highlighted in the bar graphs (e.g. Figure 3) are outliers. However, unless these outliers were excluded from statistical analyses or otherwise treated differently, there is not need for highlighting them. I would recommend displaying all the data points in the bar graphs instead.
- Figure 3 – Please provide pairwise comparisons between groups along with an overall analysis. Please use lines or brackets to denote these pairwise comparisons. Currently it is not possible to tell which groups are significantly different from which ones.
- Figure 6 – Please use a line or bracket to indicate that the significant difference exists only between the implant and freeflap group. In general please use these appropriate symbols to denote pairwise comparisons wherever possible.
- I am curious as to what the differences were between the three breast reconstruction modalities in anemic patients. Did the autologous reconstruction have more drainage and blood loss compared to other methods considering the increased surgery times?
- Line 257 – “extend” to “extent”
- The authors mention in the discussions that they found no effect of age, weight, and smoking habits on drainage fluid loss etc. However, there is no data in the results to support this claim. The authors should include analyses involving these factors to show that they do not affect the outcomes. A correlation analysis, either grouped by reconstruction techniques or taken together, should suffice.
- Line 298 – This statement is incorrect. The authors show in the results that autologous reconstruction results in less fluid drainage loss.
Line 304-305 – There aren’t sufficient controls to make this statement. The best the authors can conclude here is that implants or expanders may be less risky than autologous in the context of drainage fluid loss.
Author Response
Dear Reviewer 3,
Thank you very much for your comments. We consider your comments and annotations very helpful in improving the quality of our paper and have implemented your comments. Please find our answers below:
- Thank you for this important annotation. According to your suggestion, we explained our rational for choosing to investigate the factors anemia and drainage fluid volume in detail (line 49).
- You are very right in asking for an explanation. An explanation of the term “drainage fluid volume” has been added (line 54).
- We are very sorry for choosing the wrong expression in the exclusion of the last 37 patients. The excluded patients (37 patients) were excluded due to lack of data. The sentence has been changed. These 37 patients were not included in our study because documentation about the amount of drainage fluid promoted daily was incoherent. Also, postoperative hemoglobin levels were not retrieved (line 147).
- Thank you for your annotation regarding the improvement and statistical reconsideration of our Figure 3,4,6,7: This is the default handling of outliers in SPSS but we did change it according to your suggestion to exclude the outliners. The outliners did not influence our statistical analyses. The changed graphics have been provided with a short explanation to facilitate its understanding.
- Thank you for the comment. A pairwise comparison between the groups has been added including a graph showing our calculations (figure 4, line 227)
- Thank you for your advice. Brackets have been added to the figures, paired with a short explanation of such.
- Thank you very much for your comment. We are pleased to hear, that our paper aroused your curiosity.
The drainage fluid output in anemic patients with free tissue reconstruction was 664ml on average, surgery duration was 08:31 hours on average and blood loss was -2.7g/l hb on average.
Drainage fluid output in definitive implant-based reconstruction was 1177.5ml on average, surgery duration was 02:25 hours on average and blood loss was -2.2g/dl hb on average.
Drainage fluid output in tissue expander-based reconstruction was 1172.9ml on average, surgery duration was 03:05 hours on average and blood loss was -2.0g/dl hb on average. Please find a line-up below
- We are sorry for this mistake. The word “extend” has been changed to “extent” (line 303).
- Thank you for pointing out, that we did not provide our future readers with this crucial information. The requested data has been added in a short paragraph, together with p-values, to justify our findings and conclusion drawn in this paper (line 314-318).
- We are deeply sorry for this mistake. The sentence has been changed accordingly to “Regarding drainage fluid loss, we encountered elevated fluid volumes in the tissue expander and definitive implant group in our analyses when comparing reconstruction types.” (line 350).
- Thank you for the explanation and highlighting the fact, that there are not sufficient controls to state this conclusion. The sentence has been changed accordingly to “Therefore, we believe that the presence of implants or expanders is less likely to increase drainage fluid output.” (line 358).
Thank you very much for your time and support in improving our paper. We hope that we addressed your comments properly. We are pleased that you followed our invitation as reviewer as we think your annotations improved the quality of our paper. We hope to further publish papers in the Journal of Clinical Medicine.
Kind regards,
Tonatiuh Flores, M.D.
Reviewer 4 Report
The authors conducted a retrospective study to evaluate potential perioperative risk factors for increased postoperative blood loss or drainage fluid volume loss in 257 breast reconstructions. They conclude that preoperative anemia is one of the most critical factors in-24 fluencing postoperative drainage fluid output and previously assumed patient-dependent risk factors did not affect drainage output. The manuscript is well written and organized. Since the cohort is very heterogenous and a matched control group is missing, it is difficult to draw that conclusion. On my opinion, this would be necessary in order to gain statistical validity.
Author Response
Dear Reviewer 4,
Thank you very much for your comment, which we appreciate very much. Please find our explanation about your important comment below:
A matched analyses could potentially offer very interesting data. In the planning of this study, we consulted a biostatistician that suggest our approach based on the data we had collected. Therefore, we chose this approach. We agree that a future study with a matched control group can offer additional insights on this interesting topic. Based on the current results of our study, we believe it still offers new insights on optimizing breast reconstruction.
We very much appreciate your interest in our work and your effort to supporting us in our publication process. It would be a pleasure, having you review further of our papers. We hope that we could answer your question and comment accordingly.
Kind regards,
Tonatiuh Flores, M.D.
Round 2
Reviewer 3 Report
I would like to thank the authors for sufficiently addressing most of my concerns about the work. There are a few minor corrections/additions still to be made, however, which I have listed below:
- Please show the significance between the groups with a bracket and a ** in the graph itself. While the information is provided in the legends, it is always desirable to have the graphs be as self-explanatory as possible.
- I think the authors might have missed the last sentence in comment 4 of my previous review. Can the authors please include all the patient data points as a scatter of dots in all the graphs, overlayed with the bars and whiskers already present? I don’t imagine this will be hard to do in SPSS. This gives the reader a quick sense of the overall distribution of the data, number of data points per group and any interesting outliers.
- I am still a little unsure about what the authors want to convey with the 7th paragraph of the discussions section (lines 535-543). Doesn’t the data show that, in fact, tissue expanders and implants can lead to more fluid drainage loss? I realize that I mistakenly mentioned in my last comment that implants may be less risky but stand corrected after looking at the data again. The authors mention that only patients with anemia show higher drainage with implants or expanders but is that what Figure 7 shows? If it does, then figure 7 legends, and the associated paragraph, should be changed to reflect that information. My understanding was that Figure 7 shows differences in reconstruction across all patients, not just anemic ones. This seems to be a case of either data missing from the results or incorrect labeling of graphs. The authors should double check to ensure that all the data used to draw conclusions are sufficiently provided in the Results section.
Author Response
Dear Reviewer 3,
Thank you again very much for your comments, which helps us to improve the manuscript. Please find our comments below:
- Thank you very much for your comment. We have now changed the graphs according to your very helpful suggestions and hope to have sufficiently addressed your comments.
- Thank you for clarification. We agree that the graphs are much more accessible. We hope that you find our graphs now adequately displayed.
- Thank you very much for your question. We want to convey our future readers, that in fact, foreign material might lead to increased drainage volume output, yet the effect of mastectomy and especially anemia must not be neglected in primary reconstructions. Figure 7 does show the amount of fluid drained daily among all patients in the three groups. We have adapted the figure legend of figure 7 and inserted a figure reference (reference to figure 7) in line 375 to better support the statement in line 373-388. In our discussion we state that anemic patients show elevated drainage volume output regardless of reconstruction type (line 386). You are right to assume, that preoperative anemic patients show significantly increased drainage volumes. For better understanding of our conclusion and to better comprehend our statistical results and findings, we adapted the state from line 378.
We thank you very much for your time and effort. We hope that you find our corrections sufficient.
Reviewer 4 Report
The authors have well adressed all reviewer comments. I think, this work merrits publication!
Author Response
Dear Reviewer 4,
Thank you very much. We are very pleased that you deem our work to be suitable for publication.
Kind regards,
Tonatiuh Flores, M.D.